# Role of Diffusion Tensor Imaging in the Diagnosis of Traumatic Axonal Injury in Individual Patients with a Concussion or Mild Traumatic Brain Injury: A Mini-Review

**DOI:** 10.3390/diagnostics12071580

**Published:** 2022-06-29

**Authors:** Sung-Ho Jang, Min-Jye Cho

**Affiliations:** Department of Physical Medicine and Rehabilitation, College of Medicine, Yeungnam University, Namku, Daegu 705-717, Korea; strokerehab@hanmail.net

**Keywords:** concussion, mild traumatic brain injury, traumatic axonal injury, diffusion tensor imaging, diffusion tensor tractography

## Abstract

Present review paper aims to understand role of diffusion tensor imaging (DTI) and diffusion tensor tractography (DTT) in diagnosis of traumatic axonal injury (TAI), induced by head trauma, in individual patients with a concussion or mild traumatic brain injury (mTBI). Precise information on presence and severity of TAI in brain is necessary for determining appropriate therapeutic strategies. Several hundred DTI-based studies have reported TAI in concussion or mTBI. Majority of these DTI-based studies have been performed in a group of patients, whereas case studies that have reported TAI in individual patients with a concussion or mTBI are fewer. Summary of these DTI-based studies for individual patients is as follows: DTI can be used as a non-invasive tool for determining presence and severity of TAI in individual patients with concussion or mTBI. However, for diagnosis of TAI in an individual patient, several conditions are required to be met: no past history of head trauma, presence of possible conditions for TAI occurrence during head trauma, development of new clinical features after head trauma, and DTI observed abnormality of a neural structure that coincides with a newly developed clinical feature. However, further studies for a more precise diagnosis of TAI in individual patients should be encouraged.

## 1. Introduction

Traumatic brain injury (TBI) is classified as a concussion when a patient suffers a loss of consciousness (LOC) for less than six hours following the trauma, whereas when LOC lasts for six hours or more, it is classified as a diffuse axonal injury [1,2]. Concussion refers to a transient reversible neurologic dysfunction of the brain without any neurological sequela or disability resulting from head trauma [3,4,5]. By contrast, mild traumatic brain injury (mTBI) is defined as a traumatic brain injury with a LOC for 30 min or less, post-traumatic amnesia for less than 24 h, and a Glasgow Coma Scale score of 13–15 [3,6]. Concussion and mTBI are not generally accompanied by brain lesions detectable on conventional brain magnetic resonance imaging (MRI) [1,2,3,4,5]. Concussion and mTBI have been used interchangeably in the clinical care of the patient population [1,2,3,4,5].

Traumatic axonal injury (TAI) is a type of TBI involving injury to axons mainly induced by indirect shearing forces during acceleration, deceleration, and rotation of the brain following head trauma [5,7,8,9,10,11,12,13]. For the last several decades since the 1960s, TAI in patients with concussion who died of other diseases has been diagnosed through a post-mortem histopathologic examination of brain tissue during autopsy [14,15]. A definitive diagnosis of TAI in live patients with concussion or mTBI has remained a challenge, due to the absence of laboratory tests and markers for confirm of TAI, as well as limitations of available imaging techniques; especially, the low resolution of conventional brain MRI [13]. After the introduction of diffusion tensor imaging (DTI) in the 1990s, several hundred DTI-based studies have reported TAI in patients with concussion or mTBI since the first study by Arfanakis et al. in 2002 [16,17,18,19,20,21,22,23,24,25,26,27,28,29,30,31,32,33,34,35,36,37,38,39,40,41,42,43,44,45,46,47,48,49,50,51,52,53,54,55,56,57,58,59,60,61,62,63,64,65,66,67,68,69,70,71,72,73,74]. Using DTI, brain lesions were also detected in various brain pathologies including cerebral infarction, vascular Parkinsonism, multiple sclerosis, Alzeimer’s disease, moderate to severe TBI and cerebral palsy, which showed normal-appearing white matter on conventional MRI [75,76,77,78,79,80].

The aim of this study is to review the diagnostic approach to diagnose TAI in individual patients with concussion or mTBI based on the previous DTI studies. The diagnosis of TAI in individual patients with concussion or mTBI is clinically important. Precise information on the presence and severity of TAI for a specific neural structure in the brain is necessary to predict prognosis and to determine appropriate therapeutic strategies including the administration of drugs, rehabilitative exercises, or use of neuromodulation techniques such as repetitive transcranial magnetic stimulation or transcranial direct current stimulation [69,71]. The majority of DTI-based studies that reported TAI in patients with concussion or mTBI have been performed as group studies involving a large number of patients, whereas only a few dozen case studies have reported TAI in individual patients with concussion or mTBI (Table 1) [16,17,18,19,20,21,22,23,24,25,26,27,28,29,30,31,32,33,34,35,36,37,38,39,40,41,42,43,44,45,46,47,48,49,50,51,52,53,54,55,56,57,58,59,60,61,62,63,64,65,66,67,68,69,70,71,72,73,74,81]. The few dozen DTI-based studies for individual patients have attempted to demonstrate TAI in the neural structures which correspond to clinical features developed after concussion or mTBI, thereby linking a specific injury to a clinical feature, enabling diagnosis [17,18,19,23,24,25,26,28,29,30,31,32,33,34,35,36,37,38,39,40,41,42,43,44,45,46,47,48,49,50,51,52,53,54,55,56,57,58,59,60,61,62,63,64,65,66,67,69,70,71,72,74]. By contrast, several researchers have developed diagnostic methods for TAI by comparing individual DTI data with those of a normal control group using specific statistics, which were obtained by whole-brain analysis [20,22,27,68]. These methods can allow a clinician to detect the presence and severity of TAI in the whole brain of an individual patient. However, these methods have limitations due to the possibility of errors that can originate from individual anatomical variations of the neural structures and the limited correlation with the clinical features of the individual patient [82]. As a result, the diagnosis of TAI of the neural structures, correlated with clinical features that develop after concussion or mTBI in individual patients is clinically useful and important.

We have reviewed the role of DTI in the diagnosis of TAI with clinical correlation in individual patients with concussion or mTBI, based on available published literature.

## 2. Method

DTI or DTT-based studies that demonstrated TAI with clinical correlations in individual patients with concussion or mTBI were searched. The electronic databases (MEDLINE database (PubMed), Web of Science, and ScienceDirect) search and manual search were used to include the case or case series studies published from 1992 until 20 May 2022. The search strategy for identifying potentially relevant articles was based on the subject heading and keywords/abbreviations with synonyms as follows: (“diffusion tensor imaging” OR “diffusion tensor tractography” OR DTI OR DTT) AND (“concussion” OR “mild traumatic brain injury” OR mTBI). The inclusion criteria for subjects were limited to studies involving individual human subjects with concussion or mTBI. Overall, 52 studies were selected to review (Figure 1) [17,18,19,23,24,25,26,28,29,30,31,32,33,34,35,36,37,38,39,40,41,42,43,44,45,46,47,48,49,50,51,52,53,54,55,56,57,58,59,60,61,62,63,64,65,66,67,69,70,71,72,74].

## 3. Comparison of Diffusion Tensor Imaging and Diffusion Tensor Tractography in the Detection of Traumatic Axonal Injury in Concussion or Mild Traumatic Brain Injury

Among various DTI or diffusion tensor tractography (DTT) parameters, the fractional anisotropy (FA) indicates the state of white matter organization and is a measure of the degree of the directionality and integrity of the white matter microstructures, the mean diffusivity (MD, or apparent diffusion coefficient): the magnitude of water diffusion, and tract volume (TV, or fiber number) is the number of voxels within a neural tract which represents the number of fibers within a neural tract [83,84,85,86,87,88]. A significant decrement of the FA or tract volume, or increment of MD compared with normal subjects, indicates injury of a neural structure or tract [83,84,85,86,87,88].

Two types of estimation methods for DTI have been commonly used to demonstrate TAI in concussion or mTBI: (1) DTI region of interest (ROI) method, which involves the measurement of DTI parameters in a certain ROI of the brain on two-dimensional DTI images, and (2) DTT method, which reconstructs the neural tracts three-dimensionally (Figure 2) [17,18,19,23,24,25,26,28,29,30,31,32,33,34,35,36,37,38,39,40,41,42,43,44,45,46,47,48,49,50,51,52,53,54,55,56,57,58,59,60,61,62,63,64,65,66,67,69,70,71,72,74,83,84,85,86,87,88]. DTI parameters such as the FA and MD can be measured in a specific neural structure using the DTI ROI method. In contrast, the DTT method allows the determination of configuration as well as DTT parameters including the FA, MD and tract volume from three-dimensionally reconstructed neural tracts [13,17,18,19,23,24,25,26,28,29,30,31,32,33,34,35,36,37,38,39,40,41,42,43,44,45,46,47,48,49,50,51,52,53,54,55,56,57,58,59,60,61,62,63,64,65,66,67,69,70,71,72,74,86,89].

In the detection of TAI in individual patients with concussion or mTBI, the DTT method has the following advantages compared to the DTI ROI method: First, the DTI ROI method can yield false results due to the high individual variation of the anatomical location of the neural structures in the human brain [82]. Second, the results of the DTI ROI method can vary depending on whether an ROI is placed in a TAI lesion (for example, a partially torn area) or a normal area (Figure 2). Third, the high inter-analyzer variability of the DTI ROI method can lead to false results [86]. By contrast, the high reliability of the DTT method has been reported [90]. The combined ROI method, which mainly used in previous studies for the DTT method, reconstructs only neural fibers passing through multiple ROI areas that are well-defined ROI locations and meet the reconstruction conditions (the threshold value of FA and the trajectory angle for termination of tracking) for each neural tract [11,13,18,23,24,25,26,28,29,30,31,32,33,34,35,36,37,38,39,40,41,42,43,44,45,46,47,48,49,50,51,52,53,54,55,56,57,58,59,60,61,62,63,64,65,66,67,69,70,71,72,74,85]. Fourth, the unique advantage of the DTT method over the DTI ROI method is that the entire neural tract can be evaluated in terms of the DTT parameters and configuration. The configuration analysis of the reconstructed neural tracts uses abnormal findings such as partial tearing, narrowing or discontinuation to detect TAI of the neural tracts in individual patients with concussion or mTBI (Figure 3) [13,18,23,24,25,26,28,29,30,31,32,33,34,35,36,37,38,39,40,41,42,43,44,45,46,47,48,49,50,51,52,53,54,55,56,57,58,59,60,61,62,63,64,65,66,67,69,70,71,72,74]. As a result, the majority of studies which have reported TAI with clinical correlations in individual patients with concussion or mTBI, have used the DTT method for various neural tracts including the spinothalamic tract, fornix, dentatorubrothalamic tract (DRTT), etc. (Figure 4) [17,18,23,24,25,26,28,29,30,31,32,33,34,35,36,37,38,39,40,41,42,43,44,45,46,47,48,49,50,51,52,53,54,55,56,57,58,59,60,61,62,63,64,65,66,67,69,70,71,72,74]. 

However, the DTT method may underestimate or overestimate the neural tracts due to regions of fiber complexity, crossing of fibers that can prevent the full reflection of the underlying fiber architecture, and incorrect analysis conditions (the threshold value of FA and the trajectory angle for the termination of tracking) [13,85,86,91,92]. Hence, experienced analyzers are essential for the precise and reliable reconstruction of the neural tracts in DTT. However, in DTT, TAI often cannot be differentiated from abnormalities due to previous head trauma, other concurrent neurological diseases, aging, or immaturity, although TAIs in concussion have some common characteristic findings [13]. A recent study reported that TAI in concussion or mTBI is characterized by its occurrence in long neural tracts and the presence of multiple injuries which are similar to diffuse axonal injuries [93]. However, TAI in concussion or mTBI showed less severe and diffuse injuries than diffuse axonal injuries [93,94]. By contrast, increased collateral branches from the affected fornix in some patients with concussion or mTBI are a unique characteristic distinct from that seen in patients with diffuse axonal injury [23,25,33,37,50,95]. The authors suggest that these differences appear to be attributed to the fact that TAI in concussion or mTBI is caused by weaker forces than those producing diffuse axonal injuries. Thus, clinicians should keep the above characteristics in mind during the diagnosis of TAI in concussion or mTBI patients. On the other hand, the optimal timing of DTI after concussion or mTBI is not clearly elucidated. Previous studies reported that anisotropy values are more frequently elevated in the acute stage after mTBI (<2 weeks), while anisotropy values are more frequently stabilized in the post-acute stage (>2 weeks) [96]. The TAI lesions in mTBI are known to persist for approximately 10 years after head trauma [97]. As a result, we think that the post-acute stage after concussion or mTBI might be more appropriate for DTI scanning to diagnose TAI. However, further studies on this topic should be encouraged. 

## 4. Diagnostic Approach to Traumatic Axonal Injuries in Individual Patients with Concussion

After the introduction of DTI, more than 50 papers have reported TAI with clinical correlations in individual patients with concussion or mTBI [17,18,19,23,24,25,26,28,29,30,31,32,33,34,35,36,37,38,39,40,41,42,43,44,45,46,47,48,49,50,51,52,53,54,55,56,57,58,59,60,61,62,63,64,65,66,67,69,70,71,72,74,81]. The diagnostic approach of the above studies to demonstrate TAI of the neural tracts can be summarized as follows: (Flowsheet 1). First, conditions of TAI compatible with the definition of concussion or mTBI are required. A brain injury, such as an acceleration–deceleration–rotation injury during head trauma should be confirmed [1,3,4,13]. On the other hand, if a patient did not experience LOC, a necessary precondition to TAI diagnosis, is an alteration in the mental state such as feeling dazed, disoriented, or confused at the time of the incident [3,13]. Second, the patient must present new clinical features after the head trauma that were not present prior to the head trauma. However, the delayed onset of the new clinical features due to a secondary axonal injury should also be considered. A secondary axonal injury is an axonal injury caused by the sequential neural injury process of an injured neural tract, even though the axons were not damaged at the time of injury itself [8,9,24,72]. Third, evidence of TAI of a neural structure on DTI or a neural tract on DTT [13,17,18,23,24,25,26,28,29,30,31,32,33,34,35,36,37,38,39,40,41,42,43,44,45,46,47,48,49,50,51,52,53,54,55,56,57,58,59,60,61,62,63,64,65,66,67,69,70,71,72,74]. TAI of a neural structure or tract can be demonstrated by abnormalities in the DTI or DTT parameters (significant decrement of the FA value or tract volume or increment of the MD value) for a neural structure or tract, and configurational abnormalities such as partial tearing, narrowing or discontinuation on DTT (Figure 2) [13]. Fourth, the newly developed clinical features and the functions of the injured neural tract must coincide. Fifth, the possibility of a DTI or DTT abnormality due to previous head trauma, concurrent neurological or psychiatric disease, aging, immaturity, or artifact of DTI or DTT should be considered [13]. Fifth, other pathologies including peripheral nerve injury, spinal cord injury, and musculoskeletal problems that can cause the clinical features should be excluded. Additionally, the improvement of a clinical feature with the specific management of an injured neural tract could be additional evidence for TAI [13]. An example would be a patient who presents with central pain due to a spinothalamic tract injury following head trauma. If the patient’s pain improves with the administration of specific drugs for central pain or repetitive transcranial magnetic stimulation for central pain, that would be additional evidence for TAI in this patient [36,65,69,71,74]. Abnormal findings which indicate an organic brain injury of the patient on nuclear medicine imaging or electrophysiological studies such as evoked potentials and electroencephalography could be additional supporting evidence for TAI in an individual patient [13].

**Flow sheet 1.** Diagnostic approach of traumatic axonal injury of a neural tract in an individual patient with concussion or mild traumatic brain injury (reprinted with modification and permission from Jang, S.H., Traumatic Brain Injury. In Tech. 2018; 137–154) [13].
Head trauma history compatible with mild TBI↓Development of new clinical symptoms and signs after head trauma↓Traumatic axonal injury findings on DTT for clinically relevant neural tracts-Configuration: tearing, narrowing or discontinuation-Significant change of DTT parameters: decreased fractional anisotropy ortract volume, or increased mean diffusivity↓R/O Previous head trauma, concurrent neurological disease, aging or artifact of DTT↓R/O Other pathologies(peripheral nerve injury, spinal cord injury, and musculoskeletal problems)↓Consider response to management for clinical symptoms↓Consider other clinical features and DTT findings of other neural tracts,↓Diagnosis of traumatic axonal injury

## 5. Criteria for the Determination of Traumatic Axonal Injury Based on the Findings of Diffusion Tensor Tractography in Individual Patients with Concussion or mTBI

After the introduction of DTI, to the best of our knowledge, all studies except one study, which analyzed the whole subcortical white matter using tract-based spatial statistics, have reported TAI with clinical correlation, using DTT for the neural tracts in individual patients with concussion or mTBI [16,17,18,19,21,23,24,25,26,28,29,30,31,32,33,34,35,36,37,38,39,40,41,42,43,44,45,46,47,48,49,50,51,52,53,54,55,56,57,58,59,60,61,62,63,64,65,66,67,69,70,71,72,74,81]. The decision confirming a diagnosis of TAI, in these studies was made based on the abnormalities of the DTT parameters or configuration. The problem in arriving at a definitive diagnosis is to define the criteria for the determination of TAI based on the findings of DTT of the neural tract. Several DTT-based studies have employed a standard deviation comparison of DTT parameters of the subjects with the values of age- and sex-matched normal control subjects [26,41,62]. A recent study evaluated the diagnostic sensitivity of TAI of the spinothalamic tract (STT) using DTT in 35 patients who presented with central pain after mTBI [74]. The authors analyzed the spinothalamic tract in terms of DTT parameters (FA and TV) and configuration (narrowing and/or tearing). In the study, 94.3% and 57.1% of patients showed values of one and two standard deviations below those of the control group on at least one and two DTT parameters, respectively. By contrast, 100% of the patients revealed configurational abnormalities (tearing, narrowing, or both) on DTT.

In 2020, Jang and Lee reported a new diagnostic approach to demonstrate TAI of the spinothalamic tract in five patients who presented with central pain following mTBI [63]. The authors based the presence of TAI on three DTT parameters (FA, MD, and fiber number) and the configuration (narrowing and tearing). Furthermore, the authors measured the area of the spinothalamic tract at the narrowed site on an axial DTI image. The data of each patient were compared to those of 12 age, sex, and handedness-matched healthy control subjects, using Bayesian statistics. (A theory in the field of statistics based on the Bayesian interpretation of probability, where probability expresses a degree of belief in an event) [98,99]. The authors found that abnormalities of the fiber number, narrowed area, and configurational abnormality (narrowing and/or partial tearing) were detected in at least one hemisphere of each patient. As a result, the authors concluded that their approach, which involves a statistical comparison using Bayesian statistics, between the DTT parameters of each patient and those of the age, sex, and handedness-matched control group, can be useful in the diagnosis of TAI of the spinothalamic tract in individual patients with mTBI.

The diagnosis of TAI based on the configuration of a neural tract on DTT involves a visual diagnosis of its configurational abnormality, which is essentially a subjective assessment. It has, however, been reported to be the most sensitive factor in the diagnosis of TAI in concussion or mTBI. In 2019, Jang and Lee reported a DTT-based diagnostic approach to TAI of the optic radiation (OR) in a patient who presented with a visual field defect on the Humphrey visual field test following mTBI (Figure 5) [62]. The authors measured the FA value and fiber number of each whole OR and four ROIs were placed on areas that showed narrowing and partial tearing based on the DTT configuration. The right OR showed narrowing, and the left OR revealed partial tearing in the posterior portion. The fiber number of the right OR was more than two standard deviations lower than the control mean. The FA values of the ROI 2 (the narrowed area of the right OR) and ROI 3 (the partially torn area of the left OR) were more than two standard deviations lower than the mean value of the control group. Thus, the authors suggested that a comparison of the values of the DTI parameters that estimated the abnormal area (narrowing or partially torn area) of a neural tract on a three-dimensionally reconstructed DTT, could be a useful technique for the detection of TAI of the OR in individual patients with mTBI.

## 6. Conclusions

This study reviewed the role of DTI in the diagnosis of TAI with clinical correlations in individual patients with concussion or mTBI. In summary, DTI for a specific neural structure or DTT for a specific neural tract can be used as a non-invasive tool for determining the presence and severity of TAI in individual patients with concussion or mTBI [17,18,19,23,24,25,26,28,29,30,31,32,33,34,35,36,37,38,39,40,41,42,43,44,45,46,47,48,49,50,51,52,53,54,55,56,57,58,59,60,61,62,63,64,65,66,67,69,70,71,72,74,81]. DTT has more advantages than DTI and the majority of studies have used DTT for diagnosis of TAI with clinical correlations in individual patients [17,18,23,24,25,26,28,29,30,31,32,33,34,35,36,37,38,39,40,41,42,43,44,45,46,47,48,49,50,51,52,53,54,55,56,57,58,59,60,61,62,63,64,65,66,67,69,70,71,72,74]. DTT for the neural tracts appears to have more advantages than DTI in the detection of TAI. However, for the diagnosis of TAI in an individual patient, several conditions should be met, which include no past history of head trauma, the presence of suitable conditions for the possibility of TAI occurrence during head trauma, the development of new clinical features after head trauma, and DTI or DTT abnormalities in a neural structure or tract, respectively, which coincide with the newly developed clinical features. The objective determination of DTI or DTT abnormalities is mandatory in the diagnosis of TAI in individual patients with concussion or mTBI. Considering previous studies on this topic, we think that the comparison of the data of DTI or DTT of an individual patient with those of age-, sex-, and handedness- (if needed) matched healthy control subjects, using special statistics that can compare the data of an individual subject with those of control groups, could be useful [98,99]. Further studies for the more precise diagnosis of TAI in individual patients with concussion or mTBI should be encouraged.

## Figures and Tables

**Figure 1 diagnostics-12-01580-f001:**
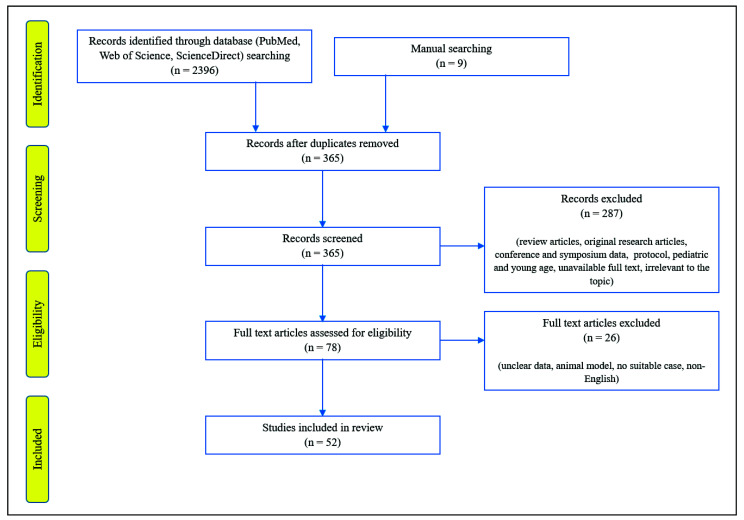
PRISMA flow diagram of search procedure.

**Figure 2 diagnostics-12-01580-f002:**
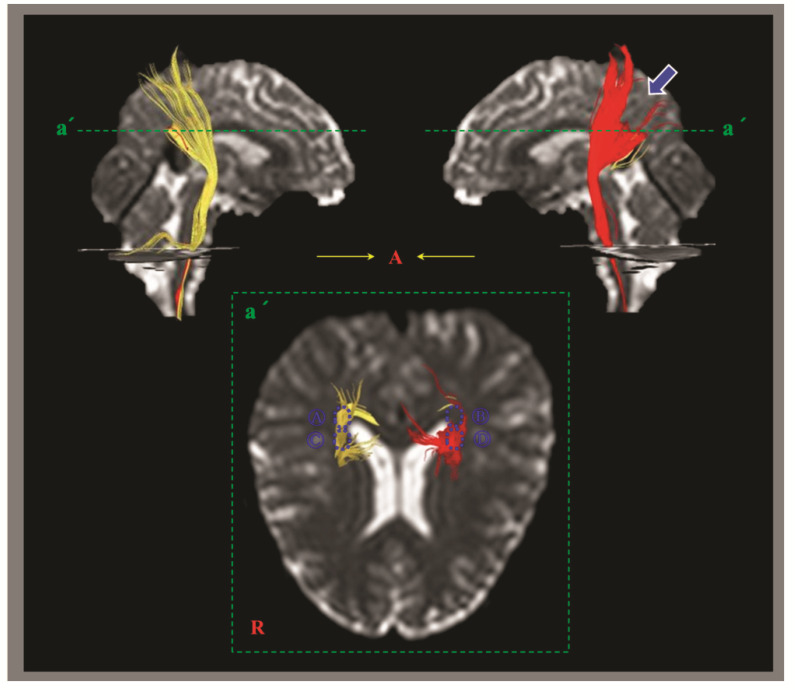
Possible false measurement of diffusion tensor imaging parameters in a partially torn corticospinal tract in a patient with mild traumatic brain injury. The left corticospinal tract shows partial tearing (arrow) at the subcortical white matter. When a researcher measures diffusion tensor imaging parameters using the region of interest (ROI) method, if the ROI is placed in the partially torn area (B), traumatic axonal injury of the left corticospinal tract can be detected, whereas if the ROI is placed in the normal-appearing area (D), traumatic axonal injury of the left corticospinal tract cannot be detected (reprinted with permission from [13]).

**Figure 3 diagnostics-12-01580-f003:**
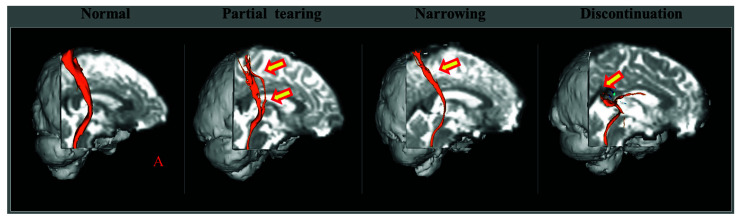
Configurational analysis of the spinothalamic tract in patients with mild traumatic brain injury. The right spinothalamic tract shows partial tearing (A), narrowing or discontinuation (arrow) (reprinted with permission from [13]).

**Figure 4 diagnostics-12-01580-f004:**
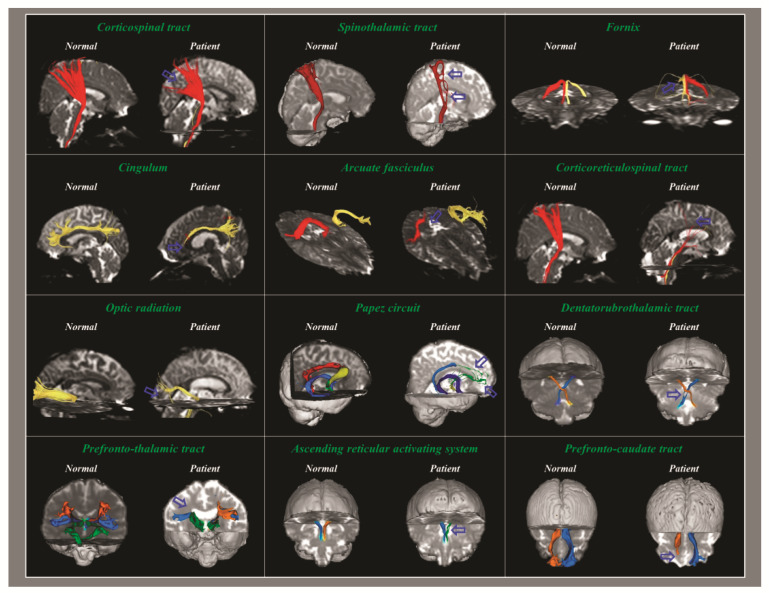
Traumatic axonal injuries (arrow) of various neural tracts (right corticospinal tract, right spinothalamic tract, right fornix, left cingulum, right arcuate fasciculus, right corticoreticulospinal tract, right optic radiation, right Papez circuit, right dentatorubrothalamic tract, right prefronto-thalamic tract, left ascending reticular activating system, right prefronto-caudate tract) in patients with concussion or mild traumatic brain injury (reprinted with permission from [12]).

**Figure 5 diagnostics-12-01580-f005:**
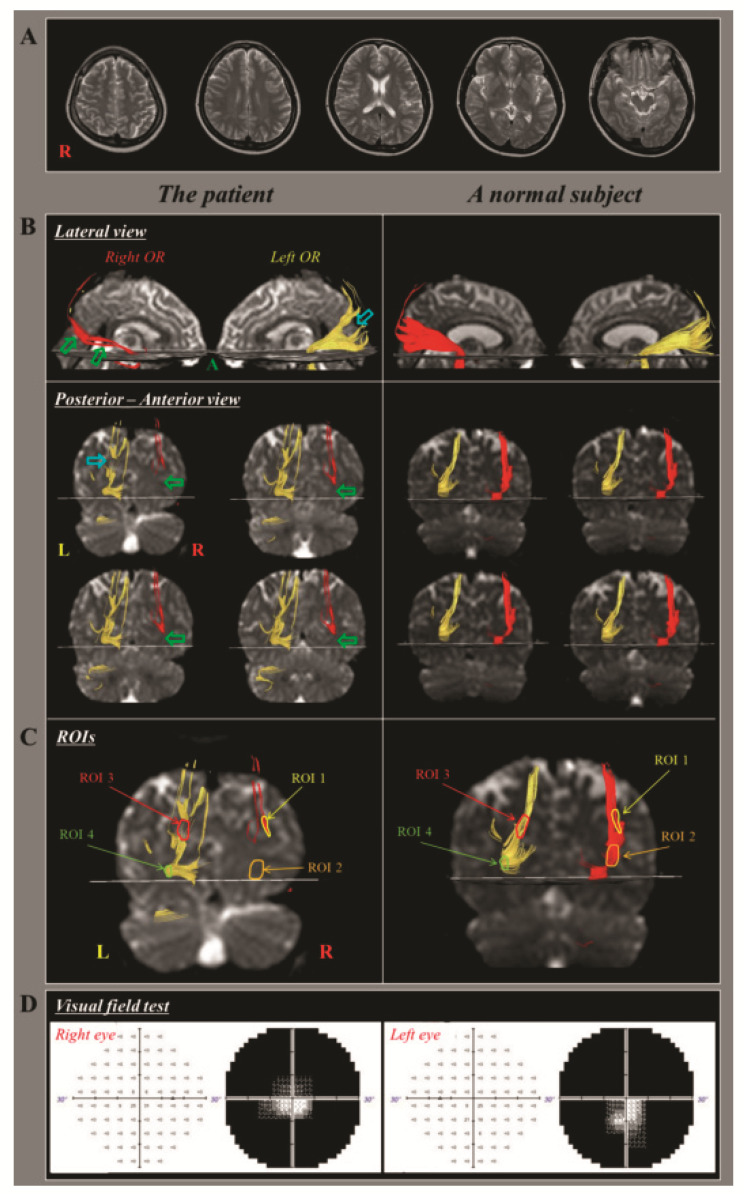
(**A**) Brain magnetic resonance images indicate no specific lesion; (**B**) Diffusion tensor tractography for the bilateral optic radiation (ORs) of the patient shows narrowing (green arrows) in the right OR and abrupt cut-off in the posterior portion (blue arrows) of the left OR compared with those in a control subject; (**C**) Four regions of interest (ROIs) in the ORs of the patient and a control subject are as follows: ROI 1, upper portion of the right OR; ROI 2, lower portion of the right ROI (narrowed area of the right OR in the patient); ROI 3, upper portion of the left OR (partially torn area of the left OR in the patient); and ROI 4, lower portion of the left ROI; (**D**) Humphrey visual field test result for the patient shows field defects in the peripheral area except for the central area of the whole visual field in both eyes (reprinted with permission [62]).

**Table 1 diagnostics-12-01580-t001:** Diffusion tensor imaging studies reporting a traumatic axonal injury in patients with a concussion or mild traumatic brain injury.

Analysed Neural Structures	Authors and Publication Year	Clinical Features
Spinothalamic tract	Seo and Jang (2014) [26] Jang and Kwon (2016) [34] Jang and Lee (2016) [36] Jang and Lee (2017) [51] Jang and Lee (2017) [52] Jang et al. (2019) [61] Jang and Lee (2019) [63] Jang and Seo (2019) [65] Jang et al. (2020) [69] Kang and Kim (2020) [71] Jang and Seo (2021) [72] Jang et al. (2022) [74]	Central pain
Fornix	Yeo and Jang (2013) [23] Lee and Jang (2014) [25] Jang and Kwon (2015) [29] Jang et al. (2016) [33] Jang and Lee (2016) [37] Jang and Lee (2017) [50] Jang and Lee (2017) [52] Jang and Seo (2017) [54]	Memory impairment
ARAS	Jang et al. (2016) [38] Jang and Kwon (2016) [35] Jang and Kwon (2017) [44] Jang and Seo (2017) [53]	Narcolepsy Fatigue and hypersomnia Daytime hypersomnia Absent-mindedness
Prefronto-thalamic tract	Jang et al. (2016) [40] Jang and Kwon (2017) [45] Jang and Kwon (2017) [46] Jang et al. (2020) [70]	Depression Akinetic mutism Apathy Cognitive dysfunction and depression
Cingulum	Kim et al. (2015) [31] Jang and Kwon (2017) [49] Jang and Lee (2017) [52] Jang et al. (2018) [57]	Memory impairment
DRTT	Jang and Kwon (2015) [28] Jang and Kwon (2017) [43] Jang and Lee (2017) [51] Chang and Seo (2020) [67]	Tremor and truncal ataxia Tremor and truncal ataxia Tremor Tremor
Corticospinal tract	Seo and Jang (2015) [32] Jang and Lee (2017) [51] Jang and Lee (2017) [52] Jang and Seo (2019) [64]	Mild weakness Mild weakness Mild weakness Hemiparesis
Papez circuit	Yang et al. (2016) [42] Jang and Kwon (2017) [47] Jang and Kwon (2018) [58]	Memory impairment
Corpus callosum	Hashimoto et al. (2007) [17] Hayes et al. (2012) [19]	Lower verbal intelligence Memory, execution, and attention impairments
Corticofugal tract	Jang and Seo (2017) [55] Jang and Seo (2017) [56]	Limb-kinetic apraxia
Corticobulbar tract	Jang and Seo (2016) [39] Jang and Lee (2018) [59]	Dysarthria Weak phonation
Optic radiation	Jang and Seo (2015) [30] Jang and Lee (2019) [62]	Visual field defect
Auditory radiation	Jang et al. (2019) [60] Lee et al. (2019) [66]	Hearing impairment Tinnitus
CRT	Kwon and Jang (2014) [24] Jang and Lee (2017) [51]	Gait disturbance Mild proximal weakness
CPCT	Jang and Kwon (2017) [48]	Tremor and ataxia
Cerebellar peduncle	Jang et al. (2016) [41]	Balance problem
Arcuate fasciculus	Rosen et al. (2009) [18]	Conduction aphasia

ARAS: ascending reticular activating system, DRTT: dentatorubrothalamic tract, CRT: corticoreticulospinal tract, CPCT: corticopontocerebellar tract.

## Data Availability

Not applicable.

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
