# Peer review of "Role of Diffusion Tensor Imaging in the Diagnosis of Traumatic Axonal Injury in Individual Patients with a Concussion or Mild Traumatic Brain Injury: A Mini-Review"

_diagnostics, 2022, doi:10.3390/diagnostics12071580_

Round 1

Reviewer 1 Report

The authors reviewed the literatures of DTI and DTT in individual patients with concussion or mild traumatic brain injury (mTBI). 

They summarized that DTI for a specific neural structure or DTT for a specific neural tract can be used as a non-invasive tool for determining the presence and severity of traumatic axonal injury (TAI) in individual patients with concussion or mTBI. 

There are several unclear points in this manuscript.

Q1 Among the demonstrated clinical features in this manuscript, are there any pathological conditions that cannot be diagnosed without DTI or DTT?

Q2 Although the flow sheet 1 ends "Diagnosis", what is diagnosed?

Q3 When is the best timing of DTI or DTT in individual patients with concussion or mTBI? 

Q4 Do the results of DTI or DTT change, vary, or fluctuate with time after concussion or mTBI?

Q5 From the view point of social economy, are plural or repeated examinations of DTI or DTT for concussion or mTBI beneficial?

The authors should emphasize the pathologies that only DTI or DTT can detect or diagnose and describe these in this manuscript.

If the authors revise this manuscript based on the above points, this manuscript will be more fascinated and attract the readers.

Reviewer 2 Report

Comments:

This is a review of use of DTI in diagnosis of traumatic axonal injury in patients with concussion and mild traumatic brain injury. The study question is of clinical importance but not a novel one. Some points need to be addressed.

1-     There has been many studies on this subject. Recently a systematic review has also been done by Tayebi et al in 2021. The authors should highlight how this article is different and adds our existing knowledge.

2-     Authors should mention about the search strategy used for this study. In absence of a clear search strategy, many articles on the subject seems to be have been missed out.

3-     The authors may Provide a PRISMA flow diagram

4-     The authors should state their aim clearly in the introduction part, so that the readers get an idea of flow of the article. The headings should be properly done.  E.g 1. Introduction.  And then the next heading is 2.1 and then 2.2.

5-     The authors should discuss about advantages of DTI vs DTT over each other.

6-     The diagnostic approach may be given a separate heading.

Round 2

Reviewer 1 Report

This manuscript has been an easy-to-understand mini-review with appropriate modifications by the authors.

This flow sheet 1 is now easier to understand.

Reviewer 2 Report

The authors have addressed the queries adequately. The article may be accepted.